# Analyzing Dynamic Surgical Workflows Through Multi-Scale Vision-Language Reasoning

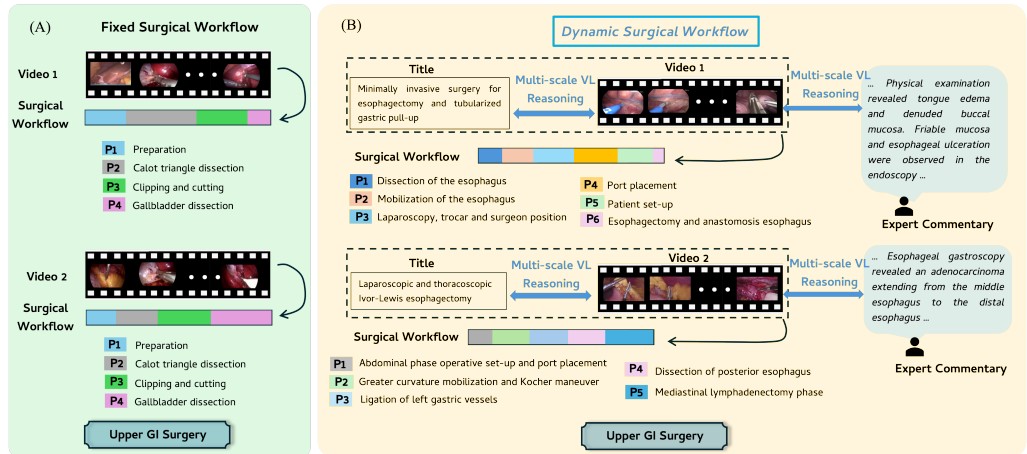

Figure 1: **Comparison between fixed surgical workflow and dynamic surgical workflow**. All the videos in the figure belong to Upper Gastrointestinal (GI) surgery. (A) In fixed surgical workflow recognition, videos are parsed into predetermined keysteps. (B) Dynamic surgical workflow reasoning aims to parse surgical videos based on the procedure's high-level objective (derived from the surgical video title), resulting in video-specific keystep identification, thus surgical keysteps would be different for different videos. We introduce the multi-scale vision-language (VL) reasoning approach grounded in expert clinical feedback to empower the reasoning ability for dynamic surgical workflow analysis.

## Abstract

Analyzing surgical workflow is critical for understanding complex procedural dynamics of a surgery. Current works focus on surgical workflow recognition to classify video streams into predetermined workflow sequences, inadequately representing the adaptive nature of clinical surgeries that respond to patient variability and evolving circumstances. We introduce the *dynamic surgical workflow reasoning* task, which eliminates fixed workflow constraints to address these limitations. Supporting this paradigm shift, we present DySurg (**Dy**namic **Surg**ical Workflow), a comprehensive dataset containing over 100 hours of long surgical videos in real-world clinical recordings with annotated dynamic workflows across 7 major surgical categories. To enhance reasoning capabilities for this analysis, we propose an commentary-aligned video reasoning framework that constructs top-down visual reasoning sequences modeling surgeons' cognitive processes. Our approach aligns visual embeddings from surgical videos with semantic information extracted from video title and expert commentary, thereby aligning explainability with the visual representations. During inference, our model maps video frames to the semantic space to generate appropriate workflows, without the need of expert commentary. Extensive evaluations on the DySurg dataset demonstrates that our approach significantly outperforms existing large vision-language models (e.g. QWen2.5-VL) and surgical-specific pre-trained models (e.g., SurgVLP) in recognizing dynamic surgical workflows. All code, data, and models will be publicly released after the review process concludes.

# 1 INTRODUCTION

Computer-assisted surgery systems enhance surgeon performance and patient safety in operating rooms (OR) Maier-Hein et al. (2017); Moglia et al. (2016). Surgical workflow analysis, a core capability of these systems, enables real-time identification of procedural phases, facilitating comprehensive understanding of complex surgical scenarios. Additionally, this analysis provides critical metrics for surgical skill assessment and enables actionable feedback for surgeons during procedures Vercauteren et al. (2019).

Existing research in surgical workflow analysis predominantly focuses on fixed, predefined workflows Ding et al. (2023); Liu et al. (2025); Zhang et al. (2024a); Yuan et al. (2024a; 2025); Honarmand et al. (2024); Yuan et al. (2024b), where surgical videos are assumed to follow a standardized sequence of steps as shown in Figure 1 (A). However, this approach fundamentally misrepresents the inherent variability and complexity of real-world clinical scenarios. In practice, surgical workflows are **highly dynamic** and **context-dependent**, varying significantly based on individual patient conditions, surgical complexities, and unexpected intraoperative challenges. Consider, for example, two minimally invasive upper gastrointestinal (GI) surgery, as shown in Figure 1 (B). While both surgeries might share a similar foundational structure, *the precise workflow can diverge substantially based on patient-specific factors*. A patient with comorbidities such as hypertension and diabetes requires surgical adaptations that differ from a patient in optimal health. Surgeons must ***dynamically*** modify their approach, potentially introducing additional preparatory steps, modifying surgical techniques, or implementing enhanced monitoring protocols to ensure patient safety. Consequently, dynamic surgical workflow analysis more accurately represents actual clinical scenarios compared to rigid workflow recognition.

To address this challenge task, we introduce *DySurg* (Dynamic Surgical Workflow), a comprehensive dataset for dynamic surgical workflow analysis. DySurg encompasses 470 real-world clinical surgical videos across 7 major surgical categories, totaling over 100 hours of content. Each video is annotated with temporal keystep descriptions, including precise start and end timestamps for each procedural segment. Additionally, we extract expert commentary from the original recordings by transcribing live commentary into text, preserving the temporal boundaries of each commentary segment.

Accurate surgical workflow analysis requires extensive medical knowledge and complex reasoning across diverse clinical scenarios. Drawing from this insight, we propose the **Multi-Scale Vision-Language Reasoning (MSVLR)** framework for dynamic surgical workflow analysis. Our approach constructs *multi-scale visual reasoning processes* for visual and semantic information. At the highest level, we conceptualize the entire surgical video as a cohesive unit directed toward a specific surgical objective, with the video title serving as the corresponding semantic anchor. Our commentary alignment module maps the comprehensive visual representation to this high-level semantic representation. Subsequently, we employ a video parser to partition the surgical video into intermediate-level segments based solely on visual features. Concurrently, we divide expert commentary into an equivalent number of segments, which represent surgeons' intermediate reasoning processes. These parallel visual and commentary segments undergo alignment through our commentary alignment module, enhancing visual representations with expert explanations. At the lowest level, we further decompose video segments into precise keystep predictions, using ground-truth keystep annotations as supervision signals to guide video parsing. During inference, since our pretrained commentary alignment module and video parser have learned to align with expert reasoning, our model can generate accurate dynamic surgical workflow predictions without requiring expert commentary input. Therefore, MSVLR models ***surgical visual reasoning processes aligned with surgeons' higher-level cognitive functions***.

We conduct extensive experiments on the *DySurg* dataset, comparing MSVLR with the most advanced large vision-language models and surgical-specific pre-trained models. Experiment results indicate that our approach surpasses existing models by a large margin in both surgical video parsing accuracy and workflow description generation quality. In summary, our major contributions are listed as follows:

- We propose a challenging dynamic surgical workflow reasoning task that more robustly represents the realistic challenges of clinical environments compared to conventional fixed surgical workflow recognition task.

- We construct the comprehensive *DySurg* dataset that includes more than 100 hours real-world clinical surgical videos with detailed expert commentary and dynamic surgical workflow annotations across 7 major surgical categories.

- We propose the novel multi-scale visual reasoning framework MSVLR to generate dynamic surgical workflow predictions with reasoning processes learned from expert commentary.

- Extensive experiments on the *DySurg* dataset demonstrate the effectiveness of MSVLR over existing large vision-language models and surgical-specific pre-trained models on the dynamic surgical workflow reasoning task.

## 2 DYNAMIC SURGICAL WORKFLOW DATASET

In this section, we first introduce the dynamic surgical workflow reasoning task, and then we show how we collect and annotate the DySurg dataset.

**Task Definition.** Dynamic surgical workflow reasoning is a challenging task to parse surgical videos based on specific surgical objectives, generating video-specific keysteps tailored to each procedure. This task presents significantly greater challenges than traditional fixed surgical workflow recognition and better

| Surgery Category | Dataset Statistics | | | | | |
|---|---|---|---|---|---|---|
| | Train/Test split | | Video Length (min) | | Keystep | |
| | Train | Test | mean | std | mean | std |
| Colorectal Surgery | 78 | 10 | 19.2 | 17.6 | 8.8 | 5.3 |
| Thoracic Surgery | 42 | 10 | 8.6 | 2.6 | 5.4 | 2.9 |
| Hepatobiliary Surgery | 80 | 10 | 10.8 | 6.5 | 7.1 | 4.2 |
| Hernia Surgery | 36 | 10 | 17.6 | 9.1 | 6 | 2.5 |
| Pediatric Surgery | 49 | 10 | 6.6 | 3.7 | 4.7 | 3.6 |
| Upper GI Surgery | 38 | 10 | 18.7 | 13.4 | 7.6 | 3.6 |
| Urologic Surgery | 82 | 10 | 15.7 | 14 | 7.2 | 6 |
| All | 400 | 70 | 13.9 | 12.4 | 6.5 | 4.6 |

Table 1: Detailed dataset statistics, including Train/Test split, Video Length and Keystep numbers.

reflects real-world clinical scenarios. Consider two minimally invasive esophagectomy procedures: one performed on a patient in good health and another on a patient with comorbidities such as hypertension and diabetes. Although the overarching surgical goal is the same, the fine-grained keysteps can differ significantly. In the latter case, surgeons must take additional precautions, introducing extra steps to mitigate risk, thereby altering the workflow. Figure 1 (B) illustrates this scenario, showing two similar surgeries with distinct keystep sequences. Traditional fixed surgical phase recognition approaches fail to account for the *dynamic clinical environment* and *patient variability* inherent in surgical practice. To address this limitation, we propose the dynamic surgical workflow reasoning task. Figure 1(B) shows two surgeries with the same surgical objective but different workflows. For example, the first workflow step (P1) in both Video1 and Video2 falls under the general "preparation" phase. However, the specific actions differ due to variations in patient conditions. Thus, while our dynamic workflows are not entirely rigid, they adhere to general surgical practices while also accounting for patient-specific factors.

**Dataset Collection, Cleaning and Quality Control.** We collected surgical videos from WebSurg[1], a premier online real-world clinical surgical video platform. All content on WebSurg undergoes rigorous peer review before publication, ensuring consistently high quality. We specifically targeted videos tagged as "surgical intervention", which include both the surgical procedures themselves and concurrent expert commentary. The videos were processed by extracting keyframes at a rate of 1 frame per second. We then applied a cleaning procedure to remove keyframes that: (1) did not contain actual surgical content (e.g., introductory lecture slides or background information), or (2) exhibited severe motion blur that compromised visual clarity. Additionally, we transcribed the expert commentary audio using Whisper Large-V3 Radford et al. (2023), a state-of-the-art automatic speech recognition (ASR) model, to capture the verbal professional insights accompanying the procedures. After that, each transcription is manually reviewed by a surgeon with the necessary expertise in the relevant surgical categories for ASR quality control.

**Annotation Process.** As illustrated in Figure 2 (A), each surgical video in our DySurg dataset includes multiple elements: the video title, sequential video keyframes, precise timestamps of distinct surgical workflows, and detailed descriptions of each workflow. We manually curated these annotations based on the surgical keystep information provided on WebSurg, carefully refining the original timestamps and descriptions to enhance their precision. To maintain clinical significance throughout the dataset, we conducted a thorough review of all keysteps, systematically excluding those with

---
[1]https://websurg.com/en

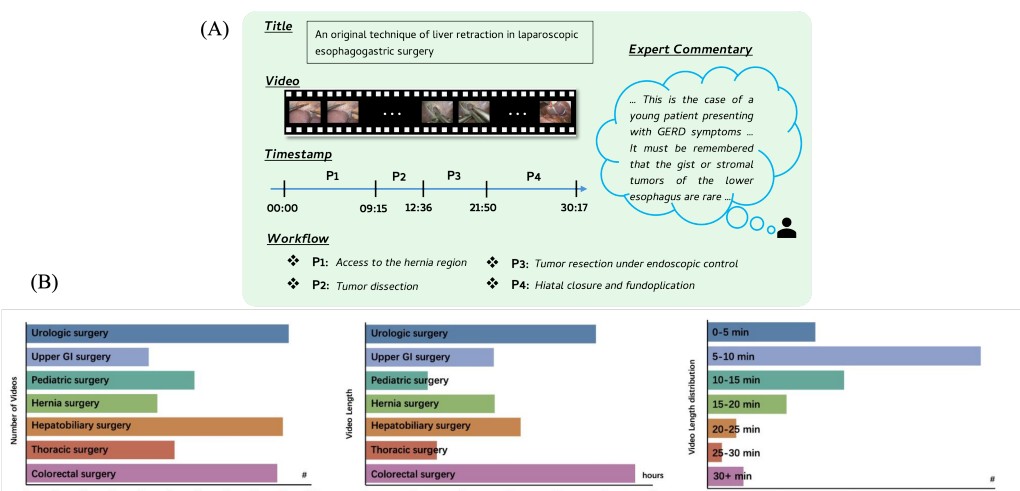

Figure 2: (A) **Example of the constructed DySurg dataset**. Each video in our dataset contains the corresponding title, dynamic workflow annotations (timestamp + workflow description) and expert commentary. (B) **Statistics of our DySurg dataset.** (1) The number of videos for each surgical category; (2) The length of each surgical category; (3) Video length distribution.

minimal clinical relevance (e.g., administrative case labeling or non-procedural content), thereby ensuring that each annotated keystep represents a meaningful surgical action or decision point.

**Dataset Statistics.** DySurg dataset consists of 470 long surgical videos, with more than 100 hours in total. The average video length is 12.4 minutes, with 6.5 keysteps. The number of videos of each surgical category, video length for each surgical category and video length distributions are shown in Figure 2 (B). Detailed dataset statistics are presented in Table 1. Ten videos from each category were designated as the test set, while the remaining videos were used for training. The average video length we collected is 13.9 minutes because longer surgeries, which can span several hours, are trimmed into concise clips by removing extended segments that depict repetitive surgical workflows. Since DySurg dataset is collected from public sources, all patient information and potentially identifiable factors have been processed to ensure anonymity. As a result, the data can be used without restriction for research purposes.

## 3 METHOD

In this section, we introduce our multi-scale vision-language reasoning framework MSVLR, which is tailored for the dynamic surgical workflow reasoning task. As shown in Figure 3, the architecture consists of three major components: (1) Video Parser for segmenting long surgical videos into keysteps; (2) Commentary-Alignment module to integrate textual commentary into visual representations; (3) Workflow text generation module to generate dynamic workflow descriptions.

### 3.1 PROBLEM FORMULATION

Given a surgical video $V = \{I_n\}_{n=1}^N$ and its title $\mathcal{T}$, where $I_n$ indicates a video keyframe, $N$ is the number of keyframes, the objective of dynamic surgical workflow reasoning is to learn a model $\mathcal{F}_\theta$ parametrized by $\theta$ for generating surgical video workflows and the descriptions of each workflow:

$$\{d_i\}_{i=1}^L, \{\mathcal{C}_i\}_{i=1}^L = \mathcal{F}_\theta(V, \mathcal{T}), \tag{1}$$

where $L$ denotes the total number of keysteps, $\{d_i\}_{i=1}^L$ indicates video workflows and $\{\mathcal{C}_i\}_{i=1}^L$ denotes the descriptions of each workflow.

### 3.2 MULTI-SCALE VISION-LANGUAGE REASONING

During surgical procedures, surgeons begin with the overall objective and systematically progressing through intermediate sub-steps to ultimately execute fine-grained surgical workflows. As demon-

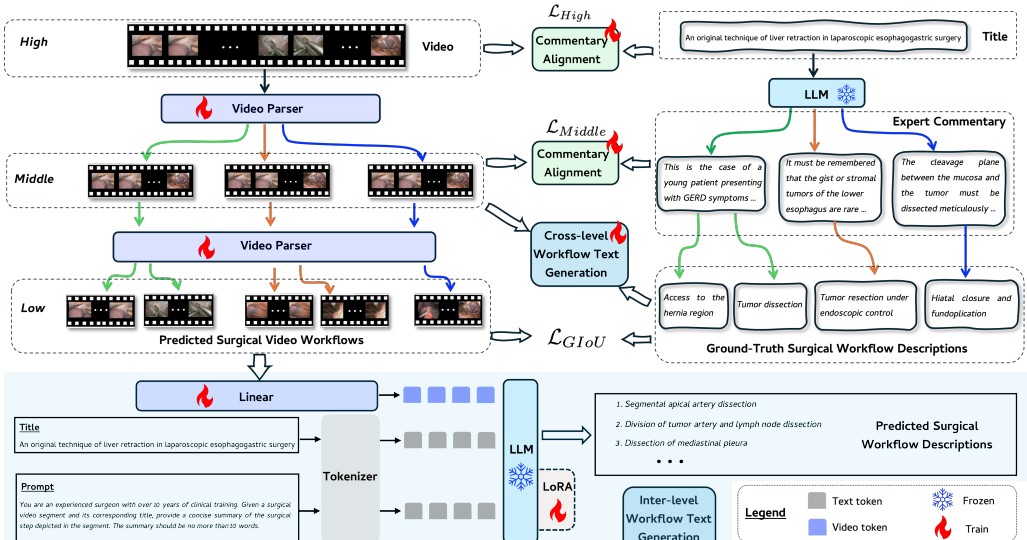

Figure 3: **Architecture of our proposed multi-scale vision-language reasoning framework.** We construct a multi-scale visual reasoning framework that integrates a commentary alignment module with video parser modules to segment surgical videos into meaningful units. On top of these representations, we design two text generation modules, inter-level and cross-level (highlighted in the blue box), to dynamically generate concise and coherent descriptions of surgical workflows.

strated in Figure 3, where the primary objective is "`liver retraction`," the surgeon strategically decomposes the procedure into distinct intermediate sub-steps, which are articulated in the accompanying expert commentary. For each of these intermediate components, the surgeon then determines and implements the most appropriate fine-grained surgical workflows specifically tailored to the particularities of the current procedure. This decision-making process reflects the adaptive reasoning that characterizes expert surgical practice.

**Dynamic Surgical Workflow Prediction.** Based on the above observations, we propose a multi-scale vision-language reasoning framework for dynamic surgical workflow reasoning. As illustrated in Figure 3, our approach establishes parallel reasoning processes between visual information (left-hand side) and language information (right-hand side). For visual processing, we initially treat the entire video as a cohesive unit. We then employ the commentary alignment module to align the holistic video representation $f_V \in \mathbb{R}^d$ with the semantic representation $f_T \in \mathbb{R}^d$ of the video title, where $d$ denotes the feature dimension. Subsequently, we deploy the video parser module to segment the complete video into intermediate sub-steps, representing the first visual reasoning transition from high to middle abstraction levels. Concurrently, we utilize a large language model (LLM) to partition the expert commentary into an equivalent number of intermediate sub-steps, maintaining structural parallelism with the visual segmentation. The commentary alignment module then integrates the contextual information from expert commentary into the corresponding visual representations. In the second visual reasoning process from the middle level to the low level, each intermediate sub-step is further decomposed into fine-grained surgical workflows. The predicted workflow sequences are directly supervised against ground-truth surgical workflow annotations. Through these two-tier visual reasoning processes, we systematically incorporate explainable expert commentary into visual representations, enhancing our model's reasoning capabilities and enabling generalization to previously unseen surgical scenarios.

**Dynamic Workflow Description Generation.** To produce concise descriptions of each identified surgical workflow, we introduce two complementary text generation modules: inter-level and cross-level workflow text generation. Both modules follow a similar pipeline, differing only in the scale of their inputs. The inter-level module operates at the most fine-grained vision–language scale, generating descriptions at a low-level granularity. In contrast, the cross-level module integrates information across the middle- and low-level scales, capturing hierarchical relationships between workflow components.

As depicted in Figure 3, we first feed the video title into the LLM alongside a carefully designed prompt. Simultaneously, we project the visual features of each predicted surgical workflow into the text embedding space using a linear projection layer, ensuring dimensional compatibility with the LLM's token representations. We then finetune the LLM using Low-Rank Adaptation (LoRA), which efficiently adapts the pretrained language model to generate accurate and clinically relevant descriptions for each surgical workflow while preserving the model's original knowledge. The loss function for LoRA finetuning is cross-entropy loss between predicted workflow descriptions and ground-truth workflow descriptions:

$$\mathcal{L}_{text} = \text{CE}(\hat{\mathcal{C}}_i, \mathcal{C}_i), \tag{2}$$

where $\hat{\mathcal{C}}_i$ denotes ground-truth workflow descriptions and $\mathcal{C}_i$ denotes predicted workflow descriptions, CE indicates cross-entropy loss. For each predicted description $\mathcal{C}_i$, its corresponding ground truth $\hat{\mathcal{C}}_i$ is selected based on the longest temporal overlap with the predicted segment.

### 3.3 VIDEO PARSER

The video parser module is designed to segment surgical videos based exclusively on visual content analysis. As illustrated in Figure 5 (A) in supplement, our approach to visual reasoning proceeds high to middle to low levels of abstraction. Initially, we partition the entire video into fixed-length video segments $S = \{I_i\}_{i=1}^{H}$, each containing $H$ consecutive keyframes. Subsequently, we employ a sliding window of length $W$ that traverses the entire video with a stride length of one segment. This sliding window mechanism facilitates the modeling of long-term temporal dependencies and inter-segment relationships, enabling the capture of contextual information that spans beyond individual segment boundaries.

In the **first visual reasoning process** from high to middle level, we split the entire video based on video segments. For video segments $\{S_i\}_{i=1}^{T}$ in a sliding window, visual representation of a video segment is computed as: $f_v^i = \frac{1}{H}\sum_{i=1}^{H} E_V(I_i) \in \mathbb{R}^d$, where $T$ denotes the number of video segments in a siding window, $E_V$ indicates a frozen visual encoder. $f_v^i$ is further updated with a transformer block containing self-attention (SA), cross-attention (CA) feed feed-forward network (FFN) in sequential: $\hat{f}_v^i = FFN(CA(K, V = SA(f_v^i); Q = \bar{f}_v)), \bar{f}_v = \frac{1}{T}\sum_{i=1}^{T} f_v^i$. To determine whether the consecutive two video segments belong to the same keystep or not, we employ a multi-layer perceptron (MLP) to generate keystep indicator $y \in (0,1)$: $y = Softmax(MLP(\hat{f}_v^t, \hat{f}_v^{t+1})), \ t \in [1, T-1]$.

For the **second visual reasoning process** from middle to low level, we determine whether the consecutive two keyframes inside a video segment belong to the same keystep. The keystep indicator $y' \in (0,1)$ is obtained from: $y' = Softmax(MLP(FFN(CA(K, V = f_{v_t}; Q = f_{v_{t+1}}))))$. By combining $y$ and $y'$, we are able to compute the generalized intersection union (GIoU) loss:

$$\mathcal{L}_{GIoU}(\hat{y}, s) = 1 - GIoU(\hat{y}, s), \tag{3}$$

$$GIoU(\hat{y}, s) = IoU - \frac{|C - \hat{y} \cup s|}{|C|}, \tag{4}$$

where $\hat{y} = [y, y']$ is the predicted video keysteps, $s$ denotes the ground-truth video keystep, $C$ is the smallest enclosing box that contains both $\hat{y}$ and $s$.

### 3.4 COMMENTARY ALIGNMENT

We introduce the commentary alignment module to integrate explainable scemantic information into visual representations. As shown in Figure 5 (B) in supplement, we take the cross attention and FFN as the main architecture of commentary alignment module.

$$f_v' = FFN(CA(K, V = f_v; Q = f_t)) \in \mathbb{R}^d, \tag{5}$$

where $f_v$ indicates visual representation generated from video parser, $f_t = E_T(\mathcal{T}) \in \mathbb{R}^d$ denotes semantic representation extracted from title or expert commentary, $E_T$ indicates a frozen text encoder. The objective of alignment loss at both high level and middle level is computed as:

$$\mathcal{L}_{High}, \mathcal{L}_{Middle} = 1 - cos\_sim(f_v', f_t), \tag{6}$$

| Method | R@0.3 | R@0.5 | R@0.7 |
|---|---|---|---|
| Qwen2.5-VL Bai et al. (2025) | 0.04 | 0.00 | 0.00 |
| UaniVTG Lin et al. (2023) | 0.16 | 0.07 | 0.03 |
| TimeChat Ren et al. (2024) | 0.09 | 0.05 | 0.03 |
| SurgVLP Yuan et al. (2025) | 0.19 | 0.11 | 0.06 |
| PeskaVLP Yuan et al. (2024a) | 0.20 | 0.09 | 0.12 |
| Video-LLM Zhang et al. (2024b) | 0.32 | 0.16 | 0.09 |
| MSVLR $\Delta$(CA) | 0.32 | 0.19 | 0.10 |
| MSVLR $\Delta$(Middle) | 0.35 | 0.28 | 0.17 |
| **MSVLR (Ours)** | **0.40** | **0.32** | **0.17** |

Table 2: **Quantitative video keystep segmentation results.** CA: Commentary Alignment; R: Recall. Results are reported at IoU thresholds 0.3, 0.5, and 0.7.

| Surgical Category | R@0.3 | R@0.5 | R@0.7 |
|---|---|---|---|
| Colorectal Surgery | 0.16 | 0.12 | 0.06 |
| Hepatobiliary Surgery | 0.69 | 0.56 | 0.39 |
| Hernia Surgery | 0.22 | 0.14 | 0.04 |
| Pediatric Surgery | 0.57 | 0.55 | 0.36 |
| Thoracic Surgery | 0.68 | 0.57 | 0.28 |
| Upper GI Surgery | 0.26 | 0.17 | 0.11 |
| Urologic Surgery | 0.14 | 0.10 | 0.04 |
| All | 0.40 | 0.32 | 0.17 |

Table 3: **Category-wise video keystep segmentation results.** Results across seven surgical categories using our commentary-aligned visual reasoning framework.

| Model | BLEU | | | METEOR | ROGUE | | | |
|---|---|---|---|---|---|---|---|---|
| | NG1 | NG2 | NG3 | | RO-1 | RO-2 | RO-L | RO-S |
| TimeChat Ren et al. (2024) | 0.0174 | 0.0027 | 0 | 0.0240 | 0.0244 | 0.0024 | 0.0244 | 0.0244 |
| SurgVLP Yuan et al. (2025) | 0.0668 | 0.0012 | 0 | 0.0395 | 0.0731 | 0.0034 | 0.0731 | 0.0731 |
| PeskaVLP Yuan et al. (2024a) | 0.0546 | 0.0009 | 0 | 0.0402 | 0.0752 | 0.0028 | 0.0752 | 0.0752 |
| Video-LLM Zhang et al. (2024b) | 0.0045 | 0 | **0.0022** | 0.0037 | 0 | 0.0037 | 0.0037 | |
| **MSVLR** $\Delta$ (CA) | 0.0487 | 0 | 0 | 0.035 | 0.0562 | 0.0030 | 0.0541 | 0.0541 |
| **MSVLR** $\Delta$ (Middle) | 0.0844 | 0.0049 | 0 | **0.0520** | 0.0809 | 0.0048 | 0.0855 | 0.0855 |
| **MSVLR (Ours)** | **0.0877** | **0.0048** | 0.0009 | 0.0491 | **0.0893** | **0.0052** | **0.0864** | **0.0864** |

Table 4: **Results on generating dynamic workflow descriptions.** CA: Commentary Alignment. NG: n_gram; RO: ROUGE.

where $cos\_sim$ denotes the cosine similarity operation.

Thus, the overall objective function for generating video segments of surgical workflows is:

$$\mathcal{L}_{seg} = \mathcal{L}_{GIoU} + \alpha\mathcal{L}_{High} + \beta\mathcal{L}_{Middle}, \tag{7}$$

where $\alpha$ and $\beta$ are scaling factor to control different loss terms.

Therefore, the overall loss function for the entire model is:

$$\mathcal{L} = \mathcal{L}_{seg} + \mathcal{L}_{text}. \tag{8}$$

To optimize the model, we adopt an interleaved training strategy: the surgical workflow segmentation module is trained for 10 epochs, followed by one epoch of training for the text generation module. This alternating process encourages the segmentation module to produce stable temporal boundaries before refining the text generation module.

## 3.5 INFERENCE

During inference, our framework takes in a surgical video and its corresponding title through the trained video parser to identify procedural keysteps. These automatically detected keysteps are then taken as input to a large language model (LLM) to generate detailed descriptive text for each step. Expert commentary is not required during inference, as our model effectively integrates learned visual representations with explainable semantic information.

## 4 EXPERIMENTS

### 4.1 IMPLEMENTATION DETAILS

We utilize ViT-B/16 from CLIP Radford et al. (2021) as the visual encoder. We employ Surgical-BERTa Bombieri et al. (2023) as text encoder, which is a language model pre-trained on surgical text. To generate keystep descriptions, we fine-tune the LLaMA-3.3 (70B) Grattafiori et al. (2024) model. For parameter-efficient fine-tuning, we incorporate LoRA Hu et al. (2022) with a rank of 8. $\alpha$ and $\beta$ in Eq 7 are both set to be 1. More implementation details are provided in the supplementary material.

| Surgical Category | BLEU | | | METEOR | ROGUE | | | |
|---|---|---|---|---|---|---|---|---|
| | NG1 | NG2 | NG3 | | RO-1 | RO-2 | RO-L | RO-S |
| Colorectal Surgery | 0.0891 | 0.0048 | 0.0011 | 0.0525 | 0.1075 | 0 | 0.1075 | 0.1075 |
| Hepatobiliary Surgery | 0.0820 | 0.0038 | 0.0009 | 0.0474 | 0.0537 | 0 | 0.0537 | 0.0537 |
| Hernia Surgery | 0.0932 | 0.0066 | 0.0020 | 0.0547 | 0.0992 | 0.0104 | 0.0913 | 0.0913 |
| Pediatric Surgery | 0.0839 | 0.0035 | 0.0008 | 0.0485 | 0.1076 | 0 | 0.1076 | 0.1076 |
| Thoracic Surgery | 0.0995 | 0.0078 | 0.0049 | 0.0442 | 0.0863 | 0.0105 | 0.0840 | 0.0840 |
| Upper GI Surgery | 0.1015 | 0.0089 | 0.0026 | 0.0514 | 0.1010 | 0.0111 | 0.0976 | 0.0976 |
| Urologic Surgery | 0.0857 | 0.0064 | 0.0015 | 0.0502 | 0.1075 | 0 | 0.1075 | 0.1075 |

Table 5: **Quantitative evaluations of dynamic workflow description generation with our MSVLR framework.** We show the detailed quantitative results of keystep description generation across 7 surgical catergories. NG: n_gram; RO: ROUGE.

## 4.2 EVALUATION METRICS

We employ Recall@1 with IoU thresholds 0.3, 0.5 and 0.7 to evaluate keystep segmentation. For keystep description generation, we employ BLEU (n_gram=1, 2,3) Papineni et al. (2002), METEOR Banerjee & Lavie (2005) and ROUGE (1, 2, L: longest common subsequence, S: sum) Lin (2004) for evaluation, which are widely used in machine translation to evaluate the quality of text from one natural language to another.

## 4.3 QUANTITATIVE RESULTS

Table 2 presents the quantitative evaluation of our approach for surgical video keystep segmentation. We benchmark our method against current state-of-the-art models for long video understanding: TimeChat Ren et al. (2024), the advanced large vision-language model Qwen2.5-VL Bai et al. (2025), surgical-specific pre-trained model SurgVLP Yuan et al. (2025) and PeskaVLP Yuan et al. (2025), as well as the leading video temporal grounding framework UniVTG Lin (2004). For Qwen2.5-VL, we implemented LoRA-based fine-tuning on our dataset. The results demonstrate that our approach substantially outperforms all baselines, achieving an R@0.3 of 0.40 compared to merely 0.04 for fine-tuned Qwen2.5-VL. Similar performance disparities are observed with TimeChat fine-tuned using LoRA, which achieves only 0.09 for R@0.3. While UniVTG, SurgVLP and PeskaVLP exhibit relatively better performance than the vision-language models, they still significantly underperform compared to our approach. These pronounced performance gaps underscore the inadequacy of generic video understanding frameworks for the specialized task of dynamic surgical keystep segmentation, highlighting the necessity of our task-specific architectural design.

For evaluating keystep description generation, we benchmark our approach against TimeChat, which offers comparable dense video captioning capabilities. We also include the surgical-specific pre-trained models SurgVLP and PeskaVLP for comparison. Although QWen2.5-VL can also process videos and generate captions, its performance is severely compromised by inadequate keystep segmentation, resulting in near-zero scores across all caption evaluation metrics. Consequently, we exclude QWen2.5-VL results from our comparative analysis in the table. Results in Table 4 indicate that our approach significantly outperforms competing emthods across all evaluation metrics. We also include reasoning-focused model (Video-LLM Zhang et al. (2024b)) for comparison. Results in Table 2 and Table 4 show that our approach consistently surpasses the reasoning-focused baseline across all evaluation metrics.

## 4.4 QUALITATIVE RESULTS

We show some qualitative results in Figure 4 of our approach. Results in Figure 4 indicate that our approach generates accurate keystep segmentation results compared with ground-truth. For keystep description generation, our approach is able to generate meaningful descriptions similar with ground-truth descriptions.

## 4.5 ABLATION STUDY

To validate the effectiveness of our proposed modules, we conduct comprehensive ablation studies by systematically removing key components. First, we evaluate the impact of the commentary alignment module by removing it, thereby eliminating semantic information and constraining the model to reason using only visual features. Second, we assess the importance of the middle-level reasoning

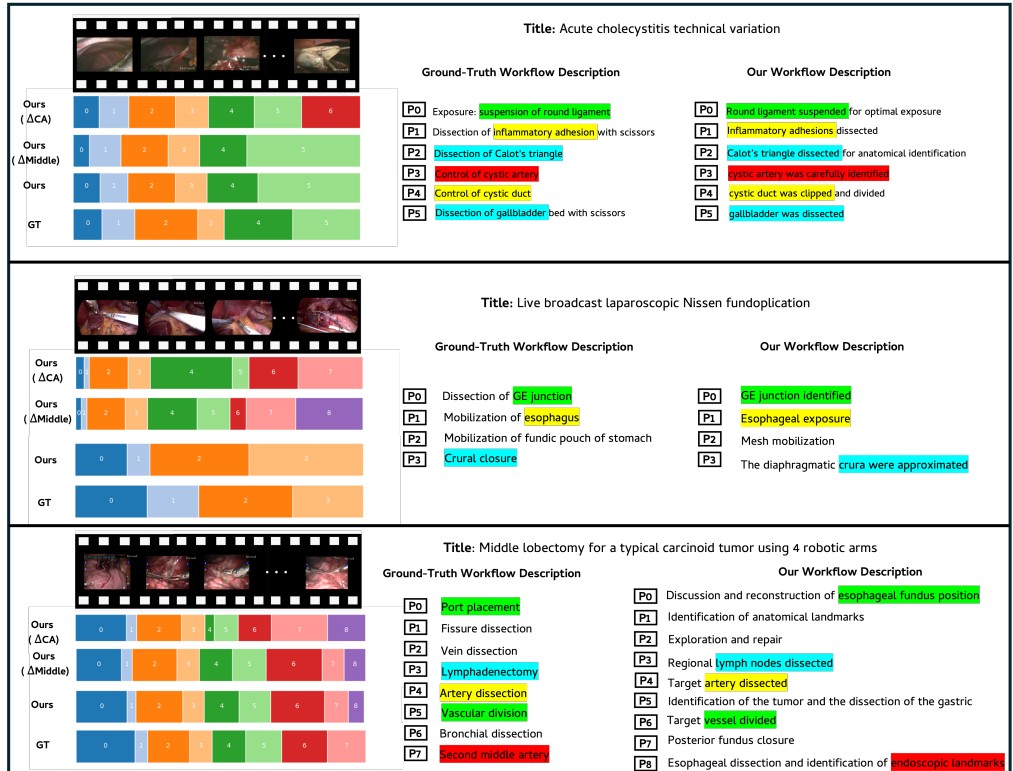

Figure 4: **Qualitative results of dynamic surgical workflow reasoning**. We present qualitative examples of dynamic surgical workflow reasoning, including video parsing results (denoted in colored boxes) and generation of keystep descriptions (labeled with $P_i$). The similar surgical notion and concepts in ground-trugh and generated workflow descriptions are highlighted with the same color for each video. CA: Commentary Alignment.

step by ablating this component, forcing the model to generate keystep segmentation results directly without the intermediate reasoning procedure. These ablations allow us to quantify the contribution of each module to the overall system performance. Results in Table 2 show that removing commentary alignment module (0.32 vs 0.40 for R@0.3) results more performance drop compared to removing the middle level reasoning procedure (0.35 vs 0.40 for R@0.3). These results indicate the semantic alignment is essential for visual reasoning process. Results in Table 4 show that removing commentary alignment module will undermine keystep description generation performance, while removing middle level reasoning procedure do not show much performance drop compared to the full model. This results is caused by the fact that the middle level reasoning procedure is mainly responsible for keystep segmentation, while for keystep description generation, its influence is limited compared to the commentary alignment module.

## 5 CONCLUSION

In this work, we address the critical limitations of conventional surgical workflow analysis by introducing a novel dynamic surgical workflow reasoning paradigm. We construct DySurg, a comprehensive dataset spanning over 100 hours of surgical videos across 7 major categories with expert-annotated dynamic workflows. In addition, we present a commentary-aligned video reasoning framework that constructs top-down reasoning sequences mimicking surgeons' cognitive processes. The proposed framework successfully aligns visual embeddings with semantic information extracted from video titles and expert commentary, enabling the model to generate appropriate keysteps without requiring expert input during inference. Extensive experimental evaluation demonstrates that our approach significantly outperforms existing large vision-language models in recognizing dynamic surgical workflows. Future work could explore extending this paradigm to other medical domains, including live commentary of surgical videos and interactive systems that provide real-time guidance during procedure.

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

**Appendix**

## A  RELATED WORK

**Surgical Workflow Recognition.** The early deep learning approaches for surgical workflow recognition include SV-RCNet Jin et al. (2017) combined ResNets for spatial feature extraction with LSTMs for temporal processing in an end-to-end framework. Gao et al. (2020) enhanced LSTM structures with tree search algorithms to leverage future context in sequence processing. Several studies future Czempiel et al. (2020); Yi et al. (2022) adopted Temporal Convolutional Networks (TCNs) for their ability to capture long-term dependencies. Models like the Anticipative Video Transformer (AVT) Girdhar & Grauman (2021) added a causal layer to ViT, working well for shorter sequences but less effectively for longer videos. TeSTra Zhao & Krähenbühl (2022) improved efficiency for long natural video sequences by selectively updating global features. The Informer model Zhou et al. (2021) reduced the computational demands of Transformers for long videos through its ProbSparse self-attention mechanism. Liu et al. (2025) extends this by combining ProbSparse with traditional self-attention—unlike previous multiscale strategiesMun et al. (2020); Fan et al. (2021) that didn't solve long sequence processing issues. Although the progress is significant, existing works on surgical workflow recognition focus on fixed surgical workflows, which neglects the dynamic nature and patient variations in clinical scenarios.

**Large Vision-Language Models for Video Understanding.** Recent advancements in Large Language Models (LLMs) have catalyzed numerous approaches that integrate LLMs with video encoders, harnessing the sophisticated comprehension and generation capabilities of LLMs for video understanding tasks Jin et al. (2024); Li et al. (2023b); Liu et al. (2023); Luo et al. (2023). These methodologies predominantly leverage open-source LLMs such as Vicuna Chiang et al. (2023) and LLaMA Touvron et al. (2023), while differing primarily in their strategies for encoding video content into vision tokens compatible with the language models. For instance, VideoChat Ren et al. (2024) employs a video transformer to encode visual features, followed by a Query Transformer (Q-Former) Li et al. (2023a) to compress video tokens into a representation suitable for language model processing. Despite demonstrating robust video understanding capabilities in natural scenes, existing large vision-language models exhibit significant limitations when applied to complex surgical scenarios. This performance gap stems primarily from the absence of high-quality, domain-specific training data tailored for the intricate reasoning demands of surgical scene analysis.

## B  DETAILED ARCHITECTURE OF VIDEO PARSER AND COMMENTARY-ALIGNMENT

## C  MORE IMPLEMENTATION DETAILS

We apply the AdamW Kingma & Ba (2014) optimizer with a learning rate of 1e-5 for training video parser and the explainability alignment module, while the LLaMA Grattafiori et al. (2024) model is optimized using SGD with a learning rate of $1e-3$. The sliding window size is 5 and each video segment contains 23 frames. The hidden dimension of embeddings in our model is 768. We finetune QWen2.5-VL (70B) Bai et al. (2025) model with LoRA to serve as one of our baseline models.

## D  MORE QUANTITATIVE RESULTS

In Table 6 and Table 7, we show the detailed results of keystep segmentation for each surgical category in two ablation conditions: (1) no middle level reasoning procedure; (2) no commentary alignment module. Results show that removing commentary alignment module results in larger performance drop than removing the middle level reasoning procedure. We also present detailed results by using the SurgVLP model Yuan et al. (2025) in Table 8, TimeChat Ren et al. (2024) model in Table 9 and by using the UniVTG Lin et al. (2023) model in Table 10.

In Table 11 and Table 12, we present detailed results of keystep description generation in the two ablation conditions same with the above.

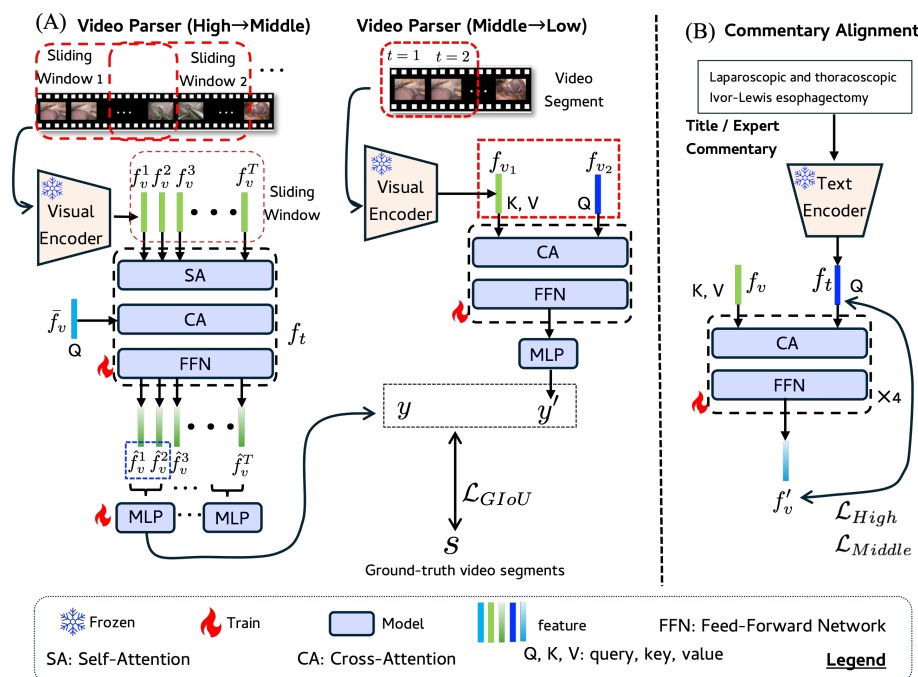

Figure 5: **Video parser and commentary alignment module**. (A) Detailed architecture of video parser for both high level to middle level and middle level to low level video segmentation. (B) Detailed architecture of commentary alignment module.

| Surgical Category | R@0.3 | R@0.5 | R@0.7 |
|---|---|---|---|
| Colorectal Surgery | 0.24 | 0.16 | 0.08 |
| Hepatobiliary Surgery | 0.68 | 0.61 | 0.45 |
| Hernia Surgery | 0.36 | 0.18 | 0.10 |
| Pediatric Surgery | 0.62 | 0.55 | 0.33 |
| Thoracic Surgery | 0.57 | 0.40 | 0.21 |
| Upper GI Surgery | 0.14 | 0.06 | 0.03 |
| Urologic Surgery | 0.26 | 0.20 | 0.16 |
| All (**MSVLR** $\Delta$ Middle) Ours | 0.35 | 0.28 | 0.18 |

Table 6: **Detailed video keystep segmentation results.** Results on 7 surgical categories with our CAR framework removing the middle level.

# E   MORE QUALITATIVE RESULTS

We show more qualitative results of dynamic surgical workflow reasoning over three different surgical categories: colorectal surgery (Figure 6), hernia surgery (Figure 7) and Thoracic surgery (Figure 8).

| Surgical Category | R@0.3 | R@0.5 | R@0.7 |
|---|---|---|---|
| Colorectal Surgery | 0.17 | 0.12 | 0.04 |
| Hepatobiliary Surgery | 0.66 | 0.42 | 0.21 |
| Hernia Surgery | 0.20 | 0.08 | 0.02 |
| Pediatric Surgery | 0.52 | 0.31 | 0.17 |
| Thoracic Surgery | 0.51 | 0.03 | 0.13 |
| Upper GI Surgery | 0.18 | 0.10 | 0.08 |
| Urologic Surgery | 0.14 | 0.10 | 0.08 |
| All (**MSVLR** $\Delta$ CA) Ours | 0.32 | 0.19 | 0.10 |

Table 7: **Detailed video keystep segmentation results.** Results on 7 surgical categories with our commentary-aligned visual reasoning framework removing the commentary alignment (CA) module.

| Surgical Category | R@0.3 | R@0.5 | R@0.7 |
|---|---|---|---|
| Colorectal Surgery | 0.08 | 0.06 | 0.05 |
| Hepatobiliary Surgery | 0.19 | 0.16 | 0.11 |
| Hernia Surgery | 0.16 | 0.08 | 0.02 |
| Pediatric Surgery | 0.40 | 0.30 | 0.14 |
| Thoracic Surgery | 0.28 | 0.13 | 0.09 |
| Upper GI Surgery | 0.10 | 0.04 | 0.01 |
| Urologic Surgery | 0.30 | 0.10 | 0.06 |
| All (**SurgVLP** Yuan et al. (2025)) | 0.19 | 0.11 | 0.06 |

Table 8: **Detailed video keystep segmentation results with SurgVLP.** Results on 7 surgical categories with the SurgVLP model.

| Surgical Category | R@0.3 | R@0.5 | R@0.7 |
|---|---|---|---|
| Colorectal Surgery | 0.13 | 0.03 | 0.00 |
| Hepatobiliary Surgery | 0.10 | 0.03 | 0.03 |
| Hernia Surgery | 0.00 | 0.00 | 0.00 |
| Pediatric Surgery | 0.00 | 0.00 | 0.00 |
| Thoracic Surgery | 0.16 | 0.16 | 0.12 |
| Upper GI Surgery | 0.09 | 0.03 | 0.00 |
| Urologic Surgery | 0.15 | 0.10 | 0.05 |
| All (**TimeChat** Ren et al. (2024)) | 0.09 | 0.05 | 0.03 |

Table 9: **Detailed video keystep segmentation results with TimeChat.** Results on 7 surgical categories with the TimeChat model.

| Surgical Category | R@0.3 | R@0.5 | R@0.7 |
|---|---|---|---|
| Colorectal Surgery | 0.13 | 0.02 | 0.00 |
| Hepatobiliary Surgery | 0.16 | 0.08 | 0.03 |
| Hernia Surgery | 0.20 | 0.04 | 0.02 |
| Pediatric Surgery | 0.17 | 0.12 | 0.02 |
| Thoracic Surgery | 0.13 | 0.08 | 0.03 |
| Upper GI Surgery | 0.12 | 0.08 | 0.05 |
| Urologic Surgery | 0.22 | 0.08 | 0.02 |
| All (**UniVTG** Lin et al. (2023)) | 0.16 | 0.07 | 0.03 |

Table 10: **Detailed video keystep segmentation results with UniVTG.** Results on 7 surgical categories with the UniVTG model.

| Surgical Category | BLEU | | | METEOR | ROGUE | | | |
|---|---|---|---|---|---|---|---|---|
| | n_gram=1 | n_gram=2 | n_gram=3 | | ROUGE-1 | ROUGE-2 | ROUGE-L | ROUGE-S |
| Colorectal Surgery | 0.0701 | 0.0074 | 0.0018 | 0.0504 | 0.0817 | 0.0119 | 0.0766 | 0.0766 |
| Hepatobiliary Surgery | 0.06 | 0 | 0 | 0.04 | 0.07 | 0 | 0.07 | 0.07 |
| Hernia Surgery | 0.06 | 0.003 | 0 | 0.04 | 0.07 | 0.002 | 0.07 | 0.07 |
| Pediatric Surgery | 0.07 | 0.003 | 0 | 0.05 | 0.07 | 0.002 | 0.07 | 0.07 |
| Thoracic Surgery | 0.07 | 0.002 | 0 | 0.03 | 0.07 | 0.004 | 0.07 | 0.07 |
| Upper GI Surgery | 0.06 | 0.004 | 0.0008 | 0.03 | 0.07 | 0.004 | 0.06 | 0.06 |
| Urologic Surgery | 0.06 | 0.004 | 0.0007 | 0.05 | 0.07 | 0.004 | 0.06 | 0.06 |

Table 11: **Quantitative evaluations of keystep description generation.** We show the detailed quantitative results of keystep description generation across 7 surgical categories with our approach removing the middle level reasoning procedure.

| Surgical Category | BLEU | | | METEOR | ROGUE | | | |
|---|---|---|---|---|---|---|---|---|
| | n_gram=1 | n_gram=2 | n_gram=3 | | ROUGE-1 | ROUGE-2 | ROUGE-L | ROUGE-S |
| Colorectal Surgery | 0.06 | 0.0031 | 0 | 0.04 | 0.07 | 0.003 | 0.06 | 0.06 |
| Hepatobiliary Surgery | 0.06 | 0.0017 | 0 | 0.04 | 0.07 | 0.002 | 0.06 | 0.06 |
| Hernia Surgery | 0.06 | 0.0012 | 0 | 0.03 | 0.06 | 0.001 | 0.06 | 0.06 |
| Pediatric Surgery | 0.06 | 0.0010 | 0 | 0.03 | 0.06 | 0.001 | 0.06 | 0.06 |
| Thoracic Surgery | 0.06 | 0.0008 | 0 | 0.04 | 0.06 | 0.001 | 0.06 | 0.06 |
| Upper GI Surgery | 0.06 | 0.0017 | 0 | 0.04 | 0.06 | 0.002 | 0.06 | 0.06 |
| Urologic Surgery | 0.06 | 0.0015 | 0 | 0.03 | 0.06 | 0.001 | 0.06 | 0.06 |

Table 12: **Quantitative evaluations of keystep description generation.** We show the detailed quantitative results of keystep description generation across 7 surgical categories with our approach removing the commentary alignment module.

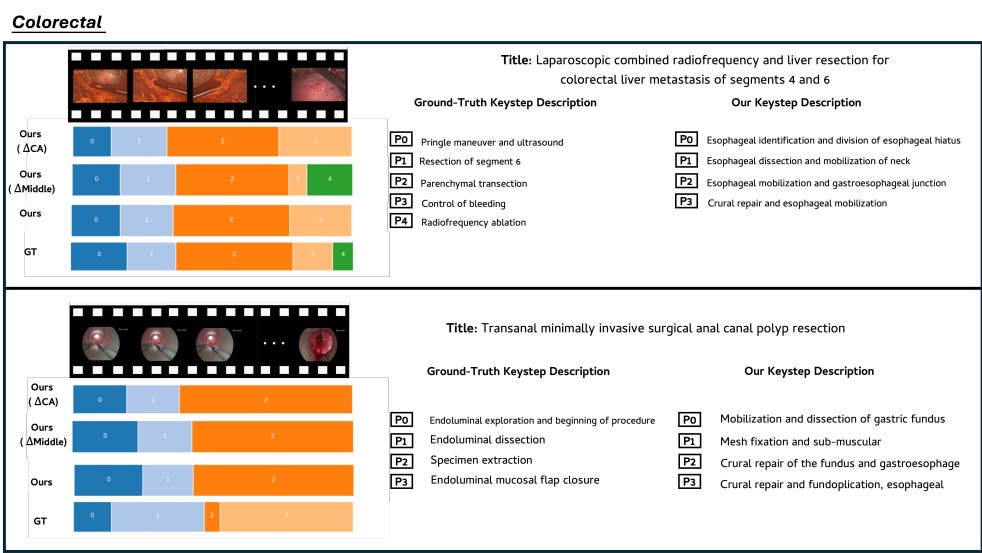

Figure 6: **Qualitative results of dynamic surgical workflow reasoning of colorectal surgery**. CA: Commentary Alignment.

*Hernia*

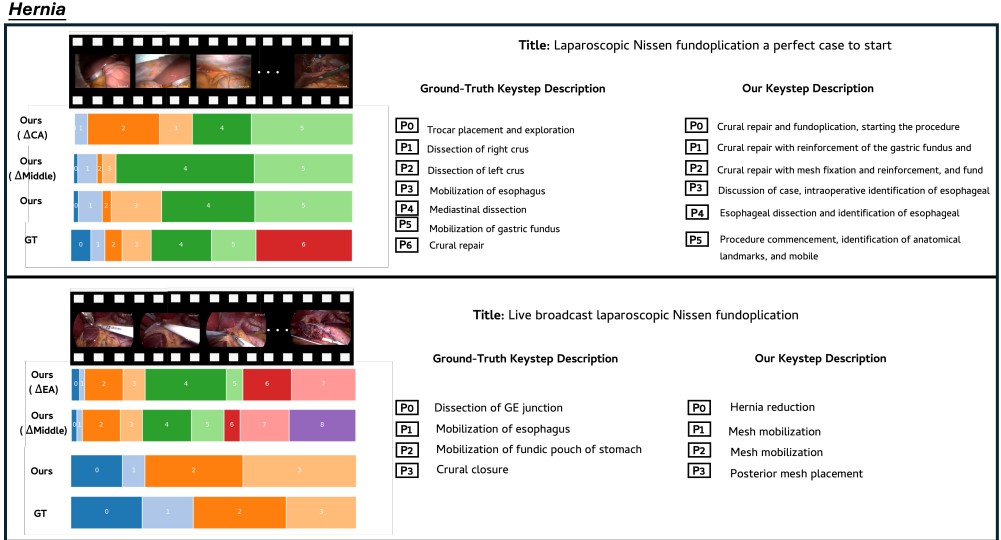

Figure 7: **Qualitative results of dynamic surgical workflow reasoning of hernia surgery**. CA: Commentary Alignment.

*Thoracic*

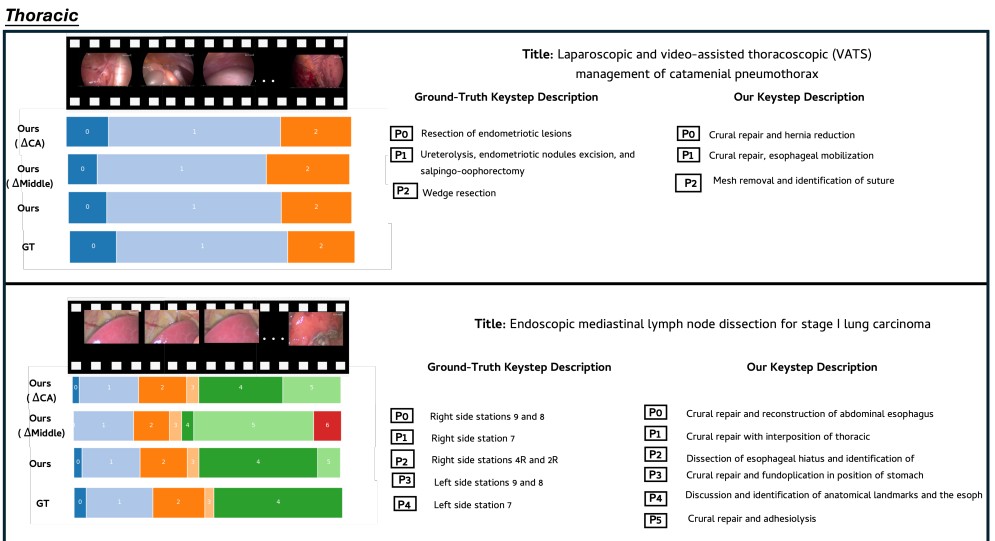

Figure 8: **Qualitative results of dynamic surgical workflow reasoning of thoracic surgery**. CA: Commentary Alignment.

