# OpenReview forum: "Analyzing Dynamic Surgical Workflows Through Multi-Scale Vision-Language Reasoning"
_ICLR.cc/2026/Conference — ICLR 2026 Conference Withdrawn Submission_

### Official Review · Reviewer_WTas · 2025-10-30

**Soundness:** 3
**Presentation:** 3
**Contribution:** 3
**Rating:** 6
**Confidence:** 4

**Summary:**

This paper proposes Dynamic Surgical Workflow Reasoning, a new paradigm replacing fixed surgical step recognition. It introduces DySurg, a 100-hour dataset of annotated surgical videos with expert commentary. The authors develop MSVLR, a multi-scale vision-language framework that aligns video and text features to model surgeons’ reasoning. MSVLR segments videos and generates step descriptions without needing expert input. Experiments show it significantly outperforms existing models in both accuracy and description quality.

**Strengths:**

1. This work proposes a novel formulation of dynamic surgical workflow reasoning, moving beyond fixed-step recognition to model real-time procedural variability, which better reflects the adaptive nature of real clinical surgeries.

2. This work introduces the DySurg dataset, a large-scale and richly annotated collection of over 100 hours of surgical videos with synchronized expert commentary, providing an unprecedented resource for visual-language learning in surgery.

3. This work develops the MSVLR framework, which uniquely aligns multi-scale visual representations with textual commentary to mimic surgeons’ hierarchical reasoning, demonstrating strong performance gains and clinical feasibility across seven surgical categories.

**Weaknesses:**

1. The proposed MSVLR framework is only evaluated on the DySurg dataset, which is collected from publicly available surgical teaching videos. These data may not accurately represent real intraoperative conditions such as occlusion, instrument clutter, or non-standard workflows. It remains unclear whether the model generalizes to live OR data or videos from different institutions.

2. Lack of comparative analysis with reasoning-focused models: Although comparisons are made with TimeChat, Qwen2.5-VL, and SurgVLP, the paper omits evaluation against reasoning-oriented video-language models such as Video-LLM (Zhang et al., 2024) or GPT-4V-based frameworks.

3. Evaluate cross-hospital or real OR data generalization, possibly through fine-tuning or zero-shot testing on other datasets (e.g., Cholec80, EndoVis).

Ref:
[1] OphNet: A Large-Scale Video Benchmark for Ophthalmic Surgical Workflow Understanding

[2] OphCLIP: Hierarchical Retrieval-Augmented Learning for Ophthalmic Surgical Video-Language Pretraining

[3] Ophora: A Large-Scale Data-Driven Text-Guided Ophthalmic Surgical Video Generation Model

[4] Towards Dynamic 3D Reconstruction of Hand-Instrument Interaction in Ophthalmic Surgery

**Questions:**

Could the authors clarify how the commentary alignment module behaves under noisy or incomplete commentary? Since real-world ASR transcripts or live commentary can be error-prone, it would be valuable to know whether the alignment is robust to missing or inaccurate text and whether any data augmentation or filtering was applied during training.

---

> ### Author Response · Authors · 2025-12-03
> **Thanks for the constructive comments. We provide our response as follows.**
>
> 1. Real world clinical relevance: All surgical videos in the DySurg dataset are sourced directly from real operating rooms and captured during actual clinical procedures. The data were collected by experienced surgeons following standard clinical protocols, ensuring high fidelity to real surgical practice. Each video additionally underwent a rigorous peer-review process to verify its clinical validity, annotation quality, and suitability for research use before public release.
>
> 2. Comparison with reasoning-focused models: We additionally compare our method with a reasoning-focused model (VideoLLM) and report the results in the tables below. The results show that our approach consistently surpasses the reasoning-focused baseline across all evaluation metrics.
>
> - Quantitative video keystep segmentation results
>
> | IOU   | R@0.3 | R@0.5 | R@0.7 |
> |:---: |:--- |:--- |:---|
> |  $$\text{VideoLLM}$$ | 0.32 | 0.16 | 0.09 |
> |  $$ \textbf{MSVLR (Ours)}$$  | **0.33** | **0.24** | **0.15** |
>
> - Results on generating dynamic workflow descriptions
>
> | BLEU   | NG1 | NG2 | BG3 |
> |:---: |:--- |:--- |:--- |
> |  $$\text{VideoLLM}$$ | 0.0045 | 0 | 0 |
> |  $$ \textbf{MSVLR (Ours)} $$ | **0.0877** | **0.0048** | **0.0009** |
>
> | ROUGE   | RG-1 | RG-2 | RG-L | RG-S
> |:---:|:--- |:--- |:--- |:--- |
> |  $$\text{VideoLLM}$$ | 0.0037 | 0 | 0.0037 | 0.0037|
> |  $$ \textbf{MSVLR (Ours)} $$ | **0.0893** | **0.0052** | **0.0864** | **0.0864**|
>
> | METEOR   |  |
> |:---:|:--- |
> |  $$\text{VideoLLM}$$ | 0.0022 |
> |  $$ \textbf{MSVLR (Ours)} $$ | **0.0491** |
>
> | Surgical Category | R@0.3 | R@0.5 | R@0.7 |
> | :--- | :---: | :---: | :---: |
> | **Urologic** | 0.54 | 0.27 | 0.13 |
> | **Hepatobiliary** | 0.26 | 0.13 | 0.07 |
> | **Colorectal** | 0.28 | 0.14 | 0.10 |
> | **Upper** | 0.39 | 0.17 | 0.14 |
> | **Hernia** | 0.20 | 0.08 | 0.02 |
> | **Pediatric** | 0.31 | 0.21 | 0.05 |
> | **Thoracic** | 0.39 | 0.17 | 0.13 |
>
> 3. Additional datasets: Our multi-scale visual reasoning framework is built upon visual–language alignment at multiple levels, including high, medium, and low. However, existing surgical workflow datasets lack detailed expert commentary, which is essential for learning such multi-level alignments. As a result, it is an unfair comparison to directly apply our framework to these existing datasets without additional expert annotations.

---

### Official Review · Reviewer_WHhH · 2025-10-30

**Soundness:** 2
**Presentation:** 2
**Contribution:** 2
**Rating:** 2
**Confidence:** 4

**Summary:**

The paper introduces the problem of „dynamic surgical workflow recognition“, where videos need to be segmented into keysteps, and the keysteps need to be described with text captions (dense video captioning). The problem specifically aims at complex and unpredictable workflows for procedures that are less standardized than the typical cholecystectomy.
To address this problem, the study compiles a new benchmark dataset („DySurg“) of 470 surgical videos across 7 major surgical categories that were previously uploaded to the WebSurg educational platform. Notably, the spoken expert commentary for each video is transcribed into text, and the text transcriptions are included in the dataset.
In addition, the work presents a method for video segmentation and captioning („MSVLR“), which leverages the transcribed expert commentary as additional signal at training time for learning high- and mid-level video representations.

**Strengths:**

1. Important and timely topic, to be useful in clinical practice, automatic methods for surgical workflow recognition need to be able to handle workflows that are more complicated than the examples in the well-known Cholec80 benchmark. This study presents a commendable step into this direction.
2. The presented DySurg dataset has the potential to serve as a novel benchmark for surgical workflow recognition on realistic, challenging cases. A big plus is the inclusion of commentary in text form.
3. The idea to integrate the available expert commentary when learning visual representations is appealing and has the potential to advance the field of vision-language modeling for surgical applications.
4. The authors promise to release all code, data, and models in the future.

**Weaknesses:**

Major
1. The problem definition and the method description lack detail and rigor to an extent that limits comprehensibility.
1a. Problem definition: The notion of a „keystep“ remains undefined. It is also not clear what a „keyframe“ is and how it differs from a common frame in the video. In line 210/211, it is not clear how „video workflows“ are defined.
1b. Video segmentation: The inputs at the lowest level (e.g. f_{v_t} in line 304/5) are undefined.
1c. Video segmentation: It is unclear how the „key step indicators“ y and y’ (at the mid and low level) are translated into a segmentation of the video. It is also not specified how the segmentation would be parameterized: the ground truth s („true video keystep“) appears in equations (3) and (4) without explanation.
1d. Text generation: It is unclear how the two modules (cross-level and inter-level workflow text generation) interact or how their predictions are fused for a final result.
2. The presented method seems inadequate for solving the problem and lacks motivation.
2a. It seems that the method is not suitable for real-time analysis of the intraoperative video stream („online recognition“), where the complete video is not available yet. In particular the mid-level segmentation, where larger portions of the video are analyzed at once, seems to violate online requirements.
2b. The method seems to repeatedly compute attention between two sequences of length 1 (i.e. individual tokens or feature vectors). This does not make much sense as the result of an individual query vector that attends over an individual key/value vector is simply the value vector itself. In combination with a residual connection, this operation basically sums the query and the value vector. What is the point?
2c. When parsing the video, it is unclear how the different levels interact and benefit from each other. In what form does information flow from top to bottom?
2d. The results are discouraging, reaching scores <0.1 on the text generation task and limited overlap with the ground truth segmentations.
3. The paper states that the presented method „models surgical visual reasoning processes aligned with surgeons’ higher-level cognitive functions“ (line 98/99) but lacks proper justification for this claim.
4. Evaluation seems superficial:
4a. The evaluation metrics should be defined in more detail.
4b. It is not clear to the reviewer why „Recall@1“ was chosen as the only metric to measure the segmentation quality.
4c. Details are missing about how the baseline models for comparison are employed, finetuned and adjusted to be applicable to what is basically a dense video captioning task. In particular, it is unclear whether the baselines were finetuned on the DySurg dataset at all.
4d. The presented MSVLR model seems to be intended to segment and describe surgical workflows of any kind. Therefore, an evaluation on common benchmarks for surgical phase recognition (Cholec80, MultiBypass140) should be included.

Minor
5. References are missing, e.g. for Transformer/attention and Generalized IoU
6. More motivation could be provided for why a natural-language description of keysteps would be required as opposed to a pre-defined categorical label. Which clinical applications are intended to be implemented based on the generated segmentation and text?
7. More information on the DySurg dataset could be provided, such as the number and frequency of different surgical steps, the number of surgeons and hospitals involved, or the variability of workflows from the same surgical category.
Limitations of the dataset should be discussed, including the limited scale (less than 100 videos per surgical category), the relatively short average duration of videos, and potential differences from ordinary intraoperative recordings, which were not edited for educational purposes.

**Questions:**

Under which conditions can WebSurg videos be used for research and be re-distributed (e.g. in form of a curated benchmark dataset)? What about the copyright of the surgeons who created the educational videos?

**Details Of Ethics Concerns:**

see questions: Under which conditions can WebSurg videos be used for research and be re-distributed (e.g. in form of a curated benchmark dataset)? What about the copyright of the surgeons who created the educational videos?

---

> ### Author Response · Authors · 2025-12-03
> **Thanks for the constructive comments. We provide our response as follows.**
>
> 1. Problem definition: A keystep represents a specific workflow segment during surgery, each corresponding to a distinct sub-surgical objective. The term video workflows refers to the ordered sequence of keysteps within a surgical video. Detailed definitions are provided in the revised paper. For keystep indicators $y$ and $y'$, video keystep predictions are obtained by combinging $y$ and $y'$. All those consecutive two video franes with keystep indicator value of 1 will be connected as the same keystep, while keystep indicator of value 0 suggest consecutive two video franes do not belong to the same keystep.
>
> 2. Methodlogy motivation:
> - Our approach is not designed for real-time analysis. Rather, our approach analyze the entire surgical video and generate surgical workflow results based on the entire video.
> - As show in Figure 5 of the Supplement Material, our method is not "repeatedly compute attention between two sequences of length 1". The sequence length is the number of frames in each video segment, which is 23 in our framework.
> - In our multi-scale visual reasoning framework, different levels interact through the video parser for the visual part and vision-language alignment.
> - The quatitaticve results are low because the task of analyzing dynamic surgical workflow is challenging. But compared with existing surgcial video foundation models and general large vision-language models, our approach is able to surpass all of them, indicating the potential of tackling this challenging task.
>
> 3. Modeling of surgical visual reasoning processes: we state that our approach "models surgical visual reasoning processes aligned with surgeons’ higher-level cognitive functions" in the sense that our model learns the hyarachical visaul reasoning processes that surgeons utilize in clinical practice. Specifically, during surgical procedures, surgeons begin with the overall objective (surgical video title) and systematically progressing through intermediate sub-steps (expert commentary) to ultimately execute fine-grained surgical workflows (ground-truth surgical workflow descriptions).
>
> 4. Evaluation results:
> - We include detailed evaluation metrics descriptions in Supplement Materials.
> - We choose Recall@1 to measure segmentation quality because only the first predicted video keystep represents our model's prediction ability.
> - In Section 4.3, we stated that all baseline models are finetuned on DySurg dataset for comparison.
> - Comparing our model on existing surgical phase recognition datasets would not be a fair comparison, since our model requires expert commentary to learn better visual representation. Exisiting surgical phase recognition datasets (Cholec80, MultiBypass140) do not contain expert commentary, whcih will lead to unfair comparison for our model.
>
> 5. Additional References: We include those references into our updated version.
>
> 6. Clincial applicaiton: One of the key clinical applications of dynamic surgical workflow analysis is medical education, particularly in the training of surgeons. By offering a detailed, step-by-step understanding of procedural phases and keysteps, this analysis supports both novice and experienced surgeons in acquiring technical skills, improving decision-making, and enhancing overall clinical proficiency.
>
> 7. More information on the DySurg dataset: We include more detailed information of DySurg dataset and discuss the limitations of the dataset in the revised paper version.

---

### Official Review · Reviewer_Z7tF · 2025-10-31

**Soundness:** 3
**Presentation:** 3
**Contribution:** 3
**Rating:** 4
**Confidence:** 5

**Summary:**

This paper introduces the novel task of dynamic surgical workflow reasoning, arguing that traditional fixed-step analysis is insufficient for real-world clinical variability. To support this new task, the authors present DySurg, a new 100+ hour dataset of surgical videos across 7 categories, annotated with dynamic workflows and aligned expert commentary. They also propose MSVLR, a multi-scale vision-language framework that aligns video features with semantic information (titles, commentary) to parse videos into procedure-specific steps. The method is shown to significantly outperform existing surgical-specific and general vision-language models on this new benchmark.

**Strengths:**

- The primary strength is the task definition. The shift from fixed to dynamic workflows is well-motivated, necessary, and highly relevant to advancing computer-assisted intervention systems for actual clinical practice.
- The DySurg dataset is a significant contribution to the field. Its size, multi-category nature, and especially the inclusion of temporally-aligned expert commentary provide a valuable resource for this new, more complex task.
- The MSVLR model's design is intuitive. The use of multi-scale reasoning (high, middle, low) combined with a commentary-alignment module to model a surgeon's cognitive process is a sound approach.

**Weaknesses:**

- The paper introduces a new dataset involving real clinical surgical videos and expert annotations. The authors state the data is from a public source, but they fail to provide a dedicated Ethics Statement addressing patient consent, data privacy, and anonymization procedures for their dataset creation and use. There is no mention of an IRB review for this new collection, which is a significant concern for any work dealing with clinical data. Additionally, a formal Reproducibility Statement detailing data, code, and model availability beyond a brief mention is absent.
- The model is trained and evaluated exclusively on the newly proposed DySurg dataset. It is unclear how this method would perform on other, existing surgical video datasets (even for fixed-phase recognition). This makes it difficult to distinguish how much of the performance gain is from the model architecture versus its specialization to the DySurg data structure.
- The method relies heavily on expert commentary during training to learn the alignment. While it is a strength that commentary is not needed for inference, acquiring such high-quality, temporally-aligned commentary for new procedures or institutions is a major practical bottleneck. This could limit the scalability and adoption of the method.
- The core idea of the MSVLR model fusing different granularities of semantic information (title, commentary) with visual features in a multi-scale manner, is a common technique in MLLM training, such as Qwen3-VL. The paper does not clearly articulate what specific architectural insights or novel components differentiate this approach from existing hierarchical fusion methods.

**Questions:**

See weaknesses

**Details Of Ethics Concerns:**

- The paper introduces a new dataset involving real clinical surgical videos and expert annotations. The authors state the data is from a public source, but they fail to provide a dedicated Ethics Statement addressing patient consent, data privacy, and anonymization procedures for their dataset creation and use. There is no mention of an IRB review for this new collection, which is a significant concern for any work dealing with clinical data. Additionally, a formal Reproducibility Statement detailing data, code, and model availability beyond a brief mention is absent.

---

> ### Author Response · Authors · 2025-12-03
> **Thanks for the constructive comments. We provide our response as follows.**
>
> 1. Ethics Statement: All surgical videos published on the WebSurg platform have undergone an IRB review process prior to public release. Furthermore, the platform’s Ethics Statement, covering patient consent, data privacy, and anonymization procedures, was reviewed and approved before the dataset became publicly available. In addition, all code, datasets, and trained models of this paper will be released upon completion of the paper’s review process.
>
> 2. Additioanl datasets: Our multi-scale visual reasoning framework is built upon visual–language alignment at multiple levels, including high, medium, and low. However, existing surgical workflow datasets lack detailed expert commentary, which is essential for learning such multi-level alignments. As a result, it is an unfair  comparison to directly apply our framework to these existing datasets without additional expert annotations.
>
> 3. Scalability of the approach: Expert commentary during surgery can serve as a generalizable component across the constructed datasets, including for new procedures or different institutions. Surgeons often provide relevant explanations or background information during operations, such as illustrating concepts or guiding junior surgeons. As long as these verbal commentaries can be collected, they can be incorporated into our dataset and leveraged by the proposed multi-scale visual reasoning framework, enabling scalable adaptation to new surgical contexts.
>
> 4. Technical novelty: The uniqueness of our multi-scale visual reasoning framework lies in its ability to explicitly model the **structure** of the reasoning process using multi-scale semantic information. While some existing MLLMs leverage different granularities of semantic features, they do not explicitly enable the model to learn the sequential and hierarchical structure of reasoning. Capturing this structure is particularly crucial for effective visual reasoning, especially in complex domains such as medicine.

---

### Official Review · Reviewer_p8te · 2025-11-01

**Soundness:** 3
**Presentation:** 3
**Contribution:** 3
**Rating:** 4
**Confidence:** 4

**Summary:**

The paper introduces DySurg, a >100-hour dataset of 470 real clinical surgical videos across 7 categories with dynamic, video-specific keystep annotations and aligned expert commentary. It formulates dynamic surgical workflow reasoning (vs. fixed-phase recognition) and proposes MSVLR, a multi-scale vision–language framework that (i) parses videos into keysteps via a two-stage visual parser, (ii) aligns visual features with title/commentary using cross-attention, and (iii) generates per-keystep textual descriptions via LoRA-tuned LLM heads (inter- and cross-level). On DySurg, MSVLR outperforms strong baselines (Qwen2.5-VL, UniVTG, TimeChat, SurgVLP/PeskaVLP) on keystep segmentation (R@0.5: 0.32) and description quality (BLEU-1: 0.088), with ablations showing the importance of commentary alignment.

**Strengths:**

Clear new task (dynamic, title-conditioned keysteps) and dataset; commentary-alignment at high/mid levels is a neat mechanistic prior for surgical reasoning.

Competitive baselines from both video-LLMs (TimeChat) and surgical VLP (SurgVLP/PeskaVLP) are included; MSVLR wins by a sizable margin on DySurg.

Method pipeline and losses are described with sufficient mathematical detail; the inference setting (no commentary) is explicit.

Addresses a limitation often noted in surveys—real surgeries are non-rigid and context-dependent—which existing fixed-phase systems underserve.

**Weaknesses:**

Training aligns frames to titles and transcribed commentary, while inference uses only titles. Without careful controls, descriptive overlap may inflate caption metrics. Recommend reporting results with title masking and commentary-drop during training and showing robustness.

DySurg is sourced from WebSurg narrated edits; real OR feeds (non-narrated, multi-view, noisy, privacy-constrained) differ substantially. Compare on non-WebSurg datasets (e.g., depth-camera OR or other phase corpora) to test transfer.

BLEU/ROUGE are low in absolute terms and may not reflect clinical usefulness; consider human evaluation (surgeons) for actionability and safety, and task-specific measures (temporal edit distance; coverage of critical safety steps).

While TimeChat/UniVTG provide temporal localization, dynamic procedure-conditioned keysteps could also be approximated by PeskaVLP/SurgVLP with retrieval+captioning; include such composed baselines to isolate the benefit of commentary alignment.

No confidence intervals or significance tests; limited per-category failure analysis (e.g., why Upper-GI underperforms). Provide CIs over multiple seeds and confusion analyses of over/under-segmentation.

**Questions:**

How were “dynamic keysteps” normalized across annotators to avoid proliferating synonyms? Any inter-rater agreement stats (κ) for boundaries and labels?

Whisper transcripts were QC’d—did you simulate residual noise or accents? How sensitive is alignment to WER perturbations?

What happens if titles are generic or misleading? Please report performance with obfuscated or shuffled titles to quantify reliance on the objective prompt.

Can the model handle multi-objective procedures (combined surgeries) or emergent events (complications)? Any plan to support branching workflows?

Any preliminary reader study with surgeons assessing whether generated keysteps are safe/complete enough for training or assistance?

---

> ### Author Response · Authors · 2025-12-03
> **Thanks for the constructive comments. We provide our response as follows.**
>
> 1. **Ablation studies**: Following the reviewers’ suggestions, we conduct an ablation study to evaluate the impact of removing the surgical video title from our framework. The results of this ablation, denoted as $\Delta\text{Title}$, are presented in the tables below:
>
> - Quantitative video keystep segmentation results
>
> | IOU   | R@0.3 | R@0.5 | R@0.7 |
> |:---: |---: |---: |---: |
> |  $$ \text{MSVLR (Ours)}\ \Delta \text{Title}$$ | 0.33 | 0.24 | 0.15 |
> |  $$ \textbf{MSVLR (Ours)} $$ | **0.40** | **0.32** | **0.17** |
>
> - Results on generating dynamic workflow descriptions
>
> | BLEU   | NG1 | NG2 | BG3 |
> |:---: |---: |---: |---: |
> |  $$ \text{MSVLR (Ours)}\ \Delta \text{Title}$$ | 0.0812 | 0.0118 | 0.0006 |
> |  $$ \textbf{MSVLR (Ours)} $$ | **0.0877** | **0.0048** | **0.0009** |
>
>
> | ROUGE   | RG-1 | RG-2 | RG-L | RG-S
> |:---: |---: |---: |---: |---: |
> |  $$ \text{MSVLR (Ours)}\ \Delta \text{Title}$$ | 0.0809 | 0.0048 | 0.0776 | 0.0776|
> |  $$ \textbf{MSVLR (Ours)} $$ | **0.0893** | **0.0052** | **0.0864** | **0.0864**|
>
>
> | METEOR   |  |
> |:---: |---: |
> |  $$ \text{MSVLR (Ours)}\ \Delta \text{Title}$$ | 0.0419 |
> |  $$ \textbf{MSVLR (Ours)} $$ | **0.0491** |
>
> - Category-wise video keystep segmentation results:
>
> | Surgical Category | R@0.3 | R@0.5 | R@0.7 |
> | :---: | ---: | ---: | ---: |
> | **Urologic** | 0.25 | 0.21 | 0.14 |
> | **Hepatobiliary** | 0.61 | 0.51 | 0.34 |
> | **Colorectal** | 0.20 | 0.13 | 0.07
> | **Upper** | 0.18 | 0.09 | 0.02 |
> | **Hernia** | 0.22 | 0.10 | 0.04 |
> | **Pediatric** | 0.36 | 0.26 | 0.19 |
> | **Thoracic** | 0.69 | 0.56 | 0.37 |
>
> For results of removing commentary, they have been included in Table 2 and Table 4 of the original paper, indicated by $\Delta (\text{CA})$.
>
> Across all experiments, removing the surgical video title leads to a consistent performance drop in both video keystep segmentation and dynamic workflow description. These findings highlight the importance of incorporating multi-scale information and further demonstrate the necessity of the multi-scale architecture in our model.
>
> 2. **Additioanl datasets**: All surgical videos in the DySurg dataset are sourced directly from real operating rooms and captured during actual clinical procedures. The data were collected by experienced surgeons following standard clinical protocols, ensuring high fidelity to real surgical practice. Each video additionally underwent a rigorous peer-review process to verify its clinical validity, annotation quality, and suitability for research use before public release.
>
> 3. **Additional evaluation metrics**: The BLEU and ROUGE scores remain relatively low due to the inherent complexity of dynamic surgical workflow analysis, which requires fine-grained temporal understanding and procedure-specific reasoning. As shown in Table 4, fine-tuning existing surgical video foundation models (SurgVLP, PeskaVLP) and video reasoning models (TimeChat) yields limited improvements and performs significantly worse than our approach. These results underscore the difficulty of the task and demonstrate that our multi-scale visual–language reasoning framework offers an effective and promising direction for addressing this challenging problem.

---

> ### Author Response · Authors · 2025-12-03
>
> (continued)
>
> 4. **Additional baselines**: In Tables 2 and 4 of the main paper, we included SurgVLP and PeskaVLP as baseline methods for comparison. For better understanding, we reorganize the relevant results below. Although dynamic, procedure-conditioned keysteps can be approximated using a retrieval-plus-captioning pipeline built upon PeskaVLP or SurgVLP, their performance remains significantly lower than that of our proposed method. These findings further highlight the limitations of existing foundation models in capturing fine-grained procedural dynamics and demonstrate the superiority of our approach.
>
> - Quantitative video keystep segmentation results
>
> | Method  | R@0.3 | R@0.5 | R@0.7 |
> | :---:  | ---: | ---: | ---: |
> | SurgVLP  | 0.19 | 0.11 | 0.06 |
> | PeskaVLP  | 0.20 | 0.09 | 0.12 |
> | **MSVLR** (Ours) | **0.40** | **0.32** | **0.17** |
>
> - Results on generating dynamic workflow descriptions
>
> | Model  | BLEU NG1 | BLEU NG2 | BLEU NG3 | METEOR | ROUGE RO-1 | ROUGE RO-2 | ROUGE RO-L | ROUGE RO-S |
> | :---:  | ---: | ---: | ---: | ---: | ---: | ---: | ---: | ---: |
> | SurgVLP  | 0.0668 | 0.0012 | 0 | 0.0395 | 0.0731 | 0.0034 | 0.0731 | 0.0731 |
> | PeskaVLP  | 0.0546 | 0.0009 | 0 | 0.0402 | 0.0752 | 0.0028 | 0.0752 | 0.0752 |
> | **MSVLR** (Ours) | **0.0877** | **0.0048** | **0.0009** | 0.0491 | **0.0893** | **0.0052** | **0.0864** | **0.0864** |
>
> 5. Confidence Intervals and Failure analysis: We provide the results of confidence intervals across multiple seeds of our model in the following tables (mean±std ).
>
> | Recall  | R@0.3 | R@0.5 | R@0.7 |
> | :--- | ---: | ---: | ---: |
> |**MSVLR** (Ours) | 0.40±0.01 | 0.30±0.02 | 0.17±0.02 |
>
> | BLEU   | NG1 | NG2 | NG3 |
> | :--- | ---: | ---: | ---: |
> | **MSVLR** (Ours) | 0.0772±0.0104 | 0.0076±0.0045 | 0.0003±0.0005 |
>
> | ROUGE | RG-1 | RG-2 | RG-L | RG-S |
> | :--- | ---: | ---: | ---: | ---: |
> | **MSVLR** (Ours)  | 0.0739±0.0134 | 0.0054±0.0002 | 0.0713±0.0131 | 0.0713±0.0131 |
>
> | METEOR | METEOR |
> | :--- | :---: |
> | **MSVLR** (Ours)  | 0.0419±0.0064 |
>
> We include a category-specific failure analysis in the updated paper. For example, the primary reason the Upper-GI category exhibits relatively lower performance is the limited number of videos available for this category, which is substantially smaller than those of the other surgical categories. This data imbalance restricts the model’s ability to learn representative patterns, leading to reduced accuracy in this subset.
>
> 6. Dynamic keysteps: All dynamic keystep annotations are sourced from the WebSurg platform, where they are created by qualified surgeons and undergo a peer-review process prior to publication. Consequently, the quality and clinical reliability of these dynamic keystep annotations are well ensured.
>
> 7. Multi-objective procedures: Handling multi-objective procedures remains highly challenging due to their complex and branching workflows. While our current model has the potential to address such cases, thanks to the flexibility of our multi-scale visual reasoning framework to incorporate additional branches, it still requires further training and carefully curated datasets that explicitly include these multi-objective examples.
>
> 8. Aablation study on video titles: As shown in the results of point 1, removing the surgical video title leads to a consistent performance drop in both video keystep segmentation and dynamic workflow description.

---

### Note · Authors · 2026-02-26

I have read and agree with the venue's withdrawal policy on behalf of myself and my co-authors.

---

### Meta-Review · Area_Chair_RUyQ · 2025-12-13

**Summary:**

This submission introduces a new task, dynamic surgical workflow reasoning, along with the DySurg dataset (470 videos, 100+ hours across 7 categories) and a multi-scale vision language framework (MSVLR) that leverages expert commentary during training and uses only titles at inference. Reviewers generally agree the task and dataset are interesting and potentially useful. Two reviewers (p8te, Z7tF) are in the borderline range (both 4) with largely positive summaries but notable reservations, one reviewer (WTas) is mildly positive (6), and one reviewer (WHhH) is clearly negative (2) with detailed concerns on problem definition, method soundness, and evaluation, plus legal and compliance questions.

After reading the paper and the rebuttal, I still lean reject. The rebuttal does address several concrete asks (title ablation, stronger baselines including VideoLLM, confidence intervals, and a brief failure analysis). However, the core concerns about rigor and clarity raised by WHhH remain substantial. In particular, the original presentation around definitions, segmentation formulation, and the multi-level design is hard to follow, and the method motivation and evaluation choices still feel under-justified. On top of that, there are unresolved compliance questions around redistribution and copyright for WebSurg content that are not convincingly cleared by a general claim that the platform had IRB and an ethics statement. Given the current level of uncertainty on both technical soundness and data usage, I recommend rejection.

**Reviewer Concerns:**

Concerns that were addressed:

1. The authors ran a title removal ablation and pointed to commentary removal results, which helps quantify reliance on metadata and commentary signals.

2. The rebuttal clarifies SurgVLP and PeskaVLP results and adds a reasoning-oriented baseline (VideoLLM).

3. Confidence intervals over multiple seeds were added, and there is some category-level discussion (Upper GI data scarcity).

Concerns that remain outstanding:

1. The negative reviewer’s point stands. Key definitions and the segmentation mechanism were not clearly presented in the original, and while the rebuttal explains them at a high level, it does not fully resolve the underlying lack of precision and readability in the technical description.

2. The justification for using Recall@1 as the primary segmentation metric remains weak, and the overall text generation scores are extremely low, which makes it hard to argue practical usefulness without stronger task-specific or human evaluation.

3. The paper is still largely evaluated on DySurg, and the argument that other datasets are “unfair” because they lack commentary does not fully address the reviewer request to test transfer or to clarify what parts of the method are expected to generalize.

4. The rebuttal addresses ethics with a brief statement, but the legal question about redistribution rights and copyright for WebSurg videos remains insufficiently supported. This is a blocking issue for a dataset paper.

**Reviewer Scores:**

Reviewer p8te: likely stays at 4. The added ablations and CIs help, but the broader concerns about evaluation quality and dataset representativeness are not fully settled.

Reviewer Z7tF: likely stays at 4. The ethics response is brief and does not fully resolve the compliance angle, and the “unfair comparison” argument does not address generalization concerns.

Reviewer WHhH: stays at 2. The reviewer raised deep issues on definition, soundness, and legal compliance, and the rebuttal does not appear to change their fundamental assessment.

Reviewer WTas: likely stays at 6. The added VideoLLM comparison helps, but the generalization and real-world deployment concerns remain.

Given the remaining technical clarity gaps and the unresolved legal and compliance risk around the dataset, my final decision is reject.

---

### Decision · Program_Chairs · 2026-01-26

Reject